# MeasureNet: Polyline Detection Based Measurement for Celiac Disease Identification and Grading

## Abstract

Celiac disease is an autoimmune disorder triggered by the consumption of gluten. It causes damage to the villi, the finger-like projections in the small intestine that are responsible for nutrient absorption. Additionally, the crypts, which form the base of the villi, are also affected, impairing the regenerative process. The deterioration in villi length, computed as the villi-to-crypt length ratio, is one indicator of the severity of the disease. However, manual measurement of villi-to-crypt length can be both time-consuming and susceptible to inter-observer variability.

Automated systems may perform measurement as a post-hoc process over segmentation outputs, but developing a villi/crypt segmentation model requires costly expert annotation, which is made worse by complex villi structures and unclear crypt boundaries. In response, our pathologists provide inexpensive annotation of (approximate) bisectors of villis/crypts for length measurement, and of auxiliary tissues that represent villi and crypt boundaries. To use such annotation, we propose MeasureNet, a polyline detection framework that integrates polyline localization with object-driven losses tailored for measurement tasks. Additionally, we leverage a segmentation model that provides auxiliary guidance on crypt locations when crypts are partially visible. To prevent over-reliance on (noisy) segmentation masks, we employ a mask feature mixup technique. We validate MeasureNet on our novel dataset CeDeM, and also a publicly available DeepBacs dataset. Compared to best SoTA baseline, MeasureNet obtains 9 and 11 percent points improvement in binary and multi-class grading of celiac disease.

## 1 Introduction

Celiac Disease (CeD) is an autoimmune disorder triggered by consuming gluten, a protein found in wheat, barley, and rye, affecting approximately 1 in 100 people worldwide (Vohra, 2016). Duodenal biopsy is a primary diagnostic tool for identifying histological changes in CeD patients. Villous atrophy, characterized by the deterioration of nutrient-absorbing villi, can lead to serious complications such as malnutrition, which can be fatal in severe cases. Pathologists assess CeD using the Q-histology score, based on IEL count, as the primary indicator (Tyagi et al., 2023) and further evaluate disease severity by calculating the villi-to-crypt length ratio (Das et al., 2019). However, this measurement process is labor-intensive and prone to significant inter- and intra-observer variability (Corazza et al., 2007). Our work focuses on developing an automated system for the prediction of tissue lengths in biopsy images. We note that length measurement has applications beyond CeD, including assessing tumor size via the length-to-width ratio (Taniyama et al., 2021) and evaluating limb alignment (Moon et al., 2023).

Segmentation of villi and crypts presents several challenges. Duodenal biopsies often feature highly disoriented or variably oriented villi, and only well-oriented, continuously sliced villi can be reliably assessed. Crypt structures, on the other hand, lack clear and consistent boundaries, making them hard to annotate. Overall, annotation for villi and crypt segmentation masks is quite time-consuming and necessitates highly trained pathologists. In discussion with pathologists, we learn that precise segmentation of villi and crypts is not critical for CeD grading, and an approximate length measurement suffices. Moreover, pathologists often rely on auxiliary signals such as villi

shoulder and crypt border (blue and yellow lines in Fig 1) to identify the limits of crypts, especially when their boundaries are unclear.

We utilize these insights to construct a pathologist-annotated training dataset of duodenal biopsies, where villi and crypts are marked by their bisector (poly)lines (green and red in the Fig 1) and additional annotation is provided for villi shoulders and crypt borders. We name this dataset, Celiac Disease Detection and Grading through Measurement (CeDeM). CeDeM comprises 752 biopsy images of the human duodenum, with over 6,800 polyline annotations for villi and crypts.

We also propose MEASURENET, which frames the task as a polyline detection problem. Its framework is built on DINO-DEtection TRansformer (DINO-DETR) (Zhang et al., 2022), and includes our novel measurement losses for measuring fine-grained geometric structures like polylines. Since crypts are

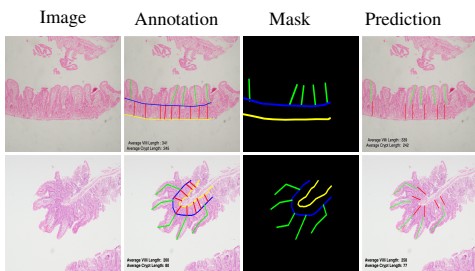

Figure 1: Column 1 shows duodenal biopsy images. Column 2 shows ground truth annotations: villi (green), crypts (red), villi shoulder (VS, blue), and crypt border (CB, yellow). Column 3 shows predicted segmentation masks, and Column 4 illustrates polyline predictions.

often not clearly visible, it needs to estimate them using projections from villi shoulder to crypt border, which are segmented using Segformer (Xie et al., 2021). The predicted segmentation masks are used as auxiliary information for detection model to learn from it. Further, to ensure that detection model does not suffer from exposure bias (Arora et al., 2022), i.e. not being exposed to mistakes during training time, we augment the segmentation mask features using mixup between strong and weak masks. To the best of our understanding, we are the first to use mixup between strong and weak segmentation masks for reducing exposure bias in training vision models.

We evaluate MEASURENET on CeDeM and also on a publicly available DeepBacs dataset (Spahn et al., 2022), used for bacterial growth measurement. Since auxiliary segmentation information is not essential for DeepBacs, this evaluates the value of our model design and proposed measurement losses. MEASURENET consistently outperforms existing approaches on both datasets. On CeDeM, MEASURENET surpasses its closest baseline by 30 percent pts in measurement, 9 pt in binary classification, and 11 pt in grade classification. On DeepBacs, it obtains 4 pt gain in measurement.

In summary: (a) We introduce the task of CeD detection through measurement. For this we contribute (and will release upon acceptance) CeDeM, a novel dataset of duodenal biopsies with polyline annotations of crypts and villi along with additional annotation of villi shoulder and crypt border. (b) We propose MEASURENET, a polyline detection framework that incorporates localization and measurement losses specifically designed for accurate measurement; (c) Guided by pathologist insights, we use segmentation masks as auxiliary information to improve crypt length and enhance feature fusion robustness through mixup; and (d) Our experiments on CeDeM and another public dataset show substantial performance improvements of MEASURENET over other existing approaches.

## 2 RELATED WORK

**Celiac Disease:** Pathologists rely on parameters like Q-histology scoring, which reflects increased intraepithelial lymphocytes (IEL), and villi-to-crypt length ratios, which indicate villous atrophy, to support CeD diagnosis (Das et al., 2019). Most AI solutions for identifying CeD use classification approaches like Carreras (2024); Püttmann et al. (2024); Sali et al. (2019), which are often non-interpretable and offer limited assistance to pathologists. DeGPR (Tyagi et al., 2023) applies the Q-histology score to classify CeD by counting IELs at villi tips but does not assess CeD severity. In this work, we focus on grading CeD via villi-crypt length measurement.

**Segmentation based Methods:** One approach to length measurement is treating it as a segmentation task, where post-processing is applied to extract lengths from predicted masks. Segmentation models like Koukoutegos et al. (2024); Xie et al. (2021); Bao et al. (2021); Ning et al. (2024); Galdran et al. (2022) are used in medical applications such as organ, tumor, and vessel segmentation. However, low confidence regions in these models can lead to fragmented predictions, introducing errors in final measurements. We compare our method against these approaches to demonstrate its effectiveness.

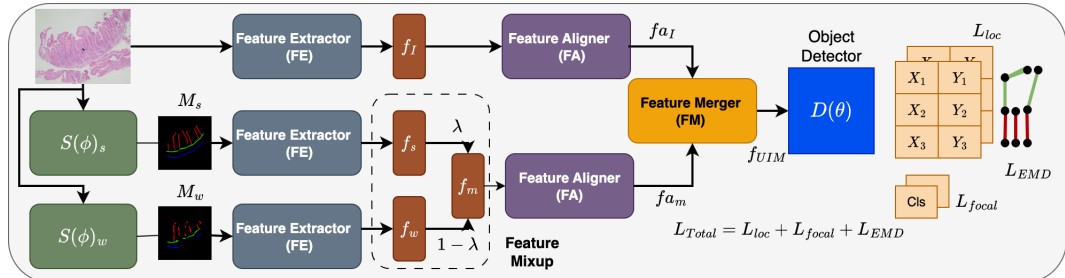

Figure 2: During training, given an input image ($I$), the segmentation models (SegFormer) $S(\phi)_s$ and $S(\phi)_w$ predict strong ($M_s$) and weak ($M_w$) segmentation masks. Features are extracted from the image ($f_I$) as well as from each mask ($f_s$ and $f_w$). Feature mixup ($f_m$) is applied before the Feature Aligner (FA) stage. The aligned image and mixed mask features are then combined using the Feature Merger (FM) module, producing a unified image-mask feature representation. These merged features are passed through the detection model $D_\theta$, a polyline detection framework, which is trained using polyline detection and measurement loss. During training $S_\phi$ and, during inference, MEASURENET utilizes only $S(\phi)_s$, eliminating the need for feature mixup.

We also employ *mixup*, which is commonly used alongside data augmentation, either in the image space (Zhang, 2017) or feature space (Wen & Li, 2021).

**Detection Methods:** Detection methods (Zhang et al., 2022) can be used for object detection however, they cannot be directly applied to polyline detection task. Curve and line detection methods have been explored in domains such as lane and map understanding. Although not originally designed for length estimation, lane detection approaches can be adapted by predicting keypoints and fitting lines. Segmentation based models like SpatialCNN (Pan et al., 2018), attention based methods such as LaneATT (Tabelini et al., 2021a;b), and single shot detectors like YoLino (Meyer et al., 2021) perform well on structured, straight lanes, but struggle with the steep curvature and irregular orientation of villi in biopsy images. Similarly, map element detection methods in autonomous driving such as VectorMapNet (Liu et al., 2022b) and MapTR (Liao et al., 2025) localize road structures effectively but fail to capture fine grained curvature, limiting their use in precise anatomical measurements. Line segment detection methods aim to identify edges or junctions and connect them into lines (Von Gioi et al., 2008; Pautrat et al., 2023; Xue et al., 2020; Teplyakov et al., 2022; Lin et al., 2024). However, they are not well suited for detecting complex anatomical structures like villi or crypts. LETR (Xu et al., 2021), a DETR based transformer model (Carion et al., 2020), predicts line segments using multi-scale features and L1 loss on endpoints. We compare MEASURENET against these methods and show consistent improvements.

## 3 MEASURENET

Our primary goal is to *measure* the lengths of villis and crypts. However, since any medical system must ideally be human verifiable, we cast it into the problem of *detection* of villi and crypts. Further, since crypts are often only partially visible, pathologists approximate crypt lengths as the distance between villi shoulder (VS) and crypt border (CB) (see Fig 1). Based on this clinical insight, we formulate our technical problem as follows: given a biopsy image $I$, output a set of polylines $P = \{P_1, P_2, \ldots, P_W\}$ with corresponding class labels $C = \{c_1, c_2, \ldots, c_n\}$ and their pixel lengths, for two classes – villi and crypts. The training data consists of polyline annotation for (approximate) bisectors of well-oriented villi and crypts, and also segmentation mask annotation for VS and CB.

Our solution, MEASURENET (see Fig. 2), is a transformer based architecture comprising segmentation $S_\phi$ plus feature extractor modules, followed by a feature integrator and detection module $D_\theta$. $S_\phi$ first generates segmentation masks for VS and CB. This is followed by feature extraction blocks for both the raw image and masks. The feature integrator module combines the aforementioned image and mask features into a unified image-mask (UIM) features for downstream polyline prediction and classification of villi and crypts. These merged features are consumed by the detection block $D_\theta$ that does the polyline prediction and classification into crypts and villi.

The detector module of MEASURENET, is based on the DINO-DETR (Zhang et al., 2022) method, albeit we introduce novel loss functions in addition to the standard localization and classification

losses. These include earth mover's distance (Rubner et al., 2000) for comparing curvature (and position), and measurement losses for comparing lengths of the predicted polylines with the ground truth. The segmentor $S_\phi$ is trained separately using auxiliary annotation, optimized with Dice and cross-entropy losses, and, additionally, a Dynamic Time Warping (DTW) loss for better alignment between the corresponding points on the VS and the CB. MEASURENET trains the modules component-wise, first training $S_\phi$ and then $D_\theta$. To reduce exposure bias (Arora et al., 2022), i.e. an over reliance on auxiliary segmentation masks, MEASURENET applies feature mixup (Zhang, 2017) on mask features obtained from a weak and a strong segmenter. The weak segmenter introduces some noise in the features, making the detector reduce its reliance on segmentation features, and thus training a more robust model. We now describe each component in detail.

### 3.1 Feature Extraction and Segmentation Modules

MEASURENET starts with extracting features from the raw image ($f_I$) and segmentation masks. We use the Swin Transformer Liu et al. (2022c) as the backbone for Feature Extractors (FE), **fine-tune with low learning rate**. For segmentation ($S_\phi$), MEASURENET uses SegFormer (Xie et al., 2021). SegFormer implements a hierarchical transformer to extract coarse-to-fine-grained features, a lightweight MLP decoder to fuse features, and these predict segmentation masks (see more details in appendix: B.2). $S_\phi$ is trained to primarily segment villi shoulder and crypt border – auxiliary information that further guides the detector (especially for crypts). We also experiment with optionally treating villi and/or crypt polylines as segmentation masks, and using $S_\phi$ to make preliminary predictions for them. We find that the best model setting is when we use villi, but not crypts (ablations in Section 6.1).

As per prevailing practice, $S_\phi$ is trained using a combination of Dice loss $L_{dice}$ and cross-entropy loss $L_{ce}$. Dice loss optimizes the overlap between predicted and ground truth regions, and the cross-entropy loss ensures pixel-wise classification accuracy. In early experiments we found that the model missed parts of VS and CB, which led to inaccurate/incomplete prediction of crypts. To ensure that each pixel in VS has a corresponding projection onto CB, $S_\phi$ additionally employs Dynamic Time Warping (DTW) (Myers et al., 2003) as a loss function ($L_{DTW}$). $L_{DTW}$ between predicted contours of VS and CB is minimum when both are aligned (details in appendix B.3). The total segmentation loss $L_{seg}$ is defined as the combination of these: $L_{\text{seg}} = L_{\text{dice}} + L_{\text{ce}} + L_{\text{DTW}}$. Once trained with $L_{seg}$, $S_\phi$ is kept frozen for training of subsequent components.

Auxiliary segmentation masks can enhance polyline detection performance; however, it may also lead to a non-robust detector. During training, segmentation masks are typically highly accurate and closely aligned with the ground truth – this may lead to detector learning to over-rely on them. In contrast, at test time, segmentation masks may contain errors, such as discontinuous predictions or false positives, which are not seen at train time (exposure bias), leading to inaccurate polyline predictions. To ensure robustness, MEASURENET employs mixup (Zhang, 2017) on mask features. Specifically, it trains two segmentation models: one trained to full capacity, referred to as strong segmentor ($S_s$), and another trained to 50% capacity – weak segmentor ($S_w$). At train time, segmentation masks from $S_s$ and $S_w$ are passed through the feature extractor to compute the mask features $f_s$ and $f_w$, respectively. These features are used to create a mixup feature representation $f_m = \lambda \cdot f_s + (1 - \lambda) \cdot f_w$. Here, $\lambda$ is sampled from a Beta distribution with hyperparameter $\delta$. Mixup ensures that MEASURENET learns to incorporate mask features with some noise, enabling it to extract meaningful information despite segmentor inaccuracies. At test time, MEASURENET only uses $S_s$ (with no mixup) for computing the segmentation-based features. To our knowledge, ours is the first application of mixup for reducing exposure bias in training computer vision models.

### 3.2 Image-Mask Feature Integrator

Once we obtain the image ($f_I$) and mask features ($f_m$), the next step is to combine them to obtain a unified representation $f_{UIM}$ (Huang et al., 2024), that goes as input to the detection block. Before merging, a Feature Aligner (FA) block is applied to align image ($f_I$) and mask ($f_m$) features, while also adapting features from the FE (trained on natural images) for histopathological images. . The Feature Aligner block uses the inverted residual block introduced in MobileNetV2 (Sandler et al.,

2018). Mathematically, the aligned features denoted by $fa_I$ and $fa_m$ are given by Eq. 1.

$$fa_I = \text{Conv}\left(\text{ReLU}\left(\text{DWConv}\left(\text{Conv}(f_I)\right)\right)\right) + (f_I)$$
$$fa_m = \text{Conv}\left(\text{ReLU}\left(\text{DWConv}\left(\text{Conv}(f_m)\right)\right)\right) + (f_m)$$

(1)

Here, DWConv represent Depth-wise convolutions.

Finally, the Feature Merger (FM) block uses a MLP-Mixer architecture (Tolstikhin et al., 2021), concatenates features $fa_I$ and $fa_m$ along the channel dimension, followed by a depth-wise convolution (Howard et al., 2017) that expands the channels to $4\times$, enabling FM to learn richer feature representations. This expanded channel is then split into two parts. One part is passed through a GELU activation, and the resulting tensors interact through a Hadamard product. This process yields $f_{UIM}$ with the same dimensionality as the original image features $f_I$. See appendix B.4 and B.5 for more details.

### 3.3 POLYLINE DETECTION

The detector $D_\theta$ adapts DINO-DETR, a transformer-based model, originally designed for bounding box detection, to predict a set of $K$ points corresponding to each polyline (more details in appendix B.1). Here, $K$ is a fixed hyperparameter, and we assume that the GT data has been post-processed to also have exactly $K$ points in every GT polyline. $D_\theta$ is trained with several loss terms relevant to the task. The first is a standard localization loss $L_{loc}$, which computes the L1 error between the point coordinates of predicted ($\hat{p}$) and ground truth ($p$) polylines: $L_{loc} = \sum_{j=1}^{K} ||p_j - \hat{p}_j||_1$ .

While individual point localization is crucial, the overall structure of the polylines formed by these points is equally important, as the predicted line $\hat{P}$ should match the curvature (and location) of the ground truth polyline $P$. For this, MEASURENET uses the Earth Mover's distance (EMD) (Rubner et al., 2000; Clason et al., 2021) as a loss function. The EMD loss is defined as:

$$L_{EMD}(P, \hat{P}) = \min_T \sum_{j=1}^{K} \sum_{k=1}^{K} T_{jk}\, d(p_j, \hat{p}_k), \text{ subject to } T_{jk} \geq 0, \sum_{k=1}^{K} T_{jk} = \frac{1}{K}, \sum_{j=1}^{K} T_{jk} = \frac{1}{K}$$

(2)

where $d(p, \hat{p})$ is defined as, $d(p, \hat{p}) = ||p - \hat{p}||_2^2$.

Since there exist instances from multiple classes with a class imbalance (many more crypts than villi), MEASURENET employs Focal Loss (Lin et al., 2020) as its classification loss. Focal Loss helps mitigate the effect of class imbalance by down-weighting the loss assigned to well-classified examples, focusing more on hard to classify instances. For polyline $P$ with class label $y$ and predicted label $\hat{y}$ for $\hat{P}$, Focal Loss is defined as: $L_{focal}(y, \hat{y}) = -\alpha(1 - \hat{y})^\gamma y \log(\hat{y})$.

For our primary objective, accurate measurement, we introduce measurement losses, Length Loss ($L_l$) and Part-Length Loss ($L_{PL}$). Length Loss ($L_L$) ensures that total length of the predicted polyline closely matches the ground truth length, defined as:

$$L_L = ||M(P) - M(\hat{P})||_2^2, \text{ where M(P)} = \sum_{j=1}^{K-1} ||p_{j+1} - p_j||_2^2$$

(3)

Similarly, the Part Length Loss ($L_{PL}$) ensures that the distances between consecutive points remain consistent, helping preserve the curvature of the polyline.

$$L_{PL} = \sum_{j=1}^{K-1} |d(p_{j+1}, p_j) - d(\hat{p}_{j+1}, \hat{p}_j)|$$

(4)

The final loss for training $D_\theta$ is $L_{final} = L_{loc} + L_{EMD} + L_{focal} + L_L + L_{PL}$. We simply add all terms, and do not introduce any hyperparameters in the final loss, to save on hyperparameter tuning.

## 4 DATASET DETAILS

We contribute (and will release) CeDeM – Celiac disease Detection and grading through Measurement – a curated dataset of 752 H&E stained duodenal histology images, annotated by expert pathol-

ogists. Images are captured using an Olympus BX50 microscope at $4\times$ magnification with a DP26 camera, producing $1920\times2148$ pixel images. CeDeM images contain 2617 villi and 4211 crypt polylines, annotated using LabelMe, along with villi shoulder (VS) and crypt border (CB) markings (see appendix A). CeDeM spans all celiac (severity) grades: 203 normal, 456 Grade 1, 75 Grade 2, and 18 Grade 3. The dataset is split into 612 training, 70 validation, and 70 test images, preserving class ratios. The 70 test images have 222 villi and 380 crypts. Each villus or crypt is annotated with 2–4 polyline points. Specifically, 1455 villi use 2 point annotations, while 997 use 3. Crypts are mostly 2 point. To standardize input for MEASURENET, we represent all structures using $K = 3$ points (start, middle, end). For 2 point annotations, the middle is interpolated; for 4 point ones, one internal point is sampled. In the dataset, villi and crypts have average lengths 104 and 47 pixels, with std deviations 25 and 15, respectively, suggesting a diverse length distribution.

We also adapt the DeepBacs dataset (Spahn et al., 2022), which provides bacterial growth segmentation masks, for length measurement. For testing our models, the style of annotation is adapted to be similar to that of CeDeM. We use 70 training, 7 validation, and 60 test images from DeepBacs. The test data has 1174 bacteria instances.

## 5 EXPERIMENTAL SETTING

### 5.1 EVALUATION METRICS

We evaluate the models along three dimensions, their localization performance, measurement accuracy, and celiac disease grade classification.

**Localization:** We evaluate localization performance using precision, recall, and mAP based on Chamfer distance (Barrow et al., 1977). A predicted polyline is considered a true positive if its Chamfer distance to a ground truth polyline is below 50. If no predicted polyline meets this threshold for a ground truth (GT) polyline, it counts as a false negative. Conversely, a predicted polyline without a matching ground truth polyline is a false positive.

**Measurement:** We evaluate the measurement quality of polylines by computing the Matched Mean Absolute Error (M-MAE). For each ground truth (GT) polyline, we identify the closest predicted polyline and compute the absolute difference (AE) between their lengths. We repeat this process for all GT polylines and compute the mean absolute difference, denoted as $\text{MAE}_{G\rightarrow P}$. Similarly, we compute $\text{MAE}_{P\rightarrow G}$ by applying the same procedure to predicted polylines. The Matched-MAE is then defined as M-MAE $= \max(\text{MAE}_{G\rightarrow P}, \text{MAE}_{P\rightarrow G})$. We use Hungarian matching based on the Chamfer distance to identify the closest matches (more details in appendix C2). Pathologists assess celiac disease by computing the average villi-to-average crypt length ratio, which we replicate by computing its Mean Absolute Error (MAE ratio).

**Celiac Classification:** The villi-to-crypt length ratio is crucial in diagnosing CeD. A ratio $> 3$ indicates a healthy patient, $1.05 - 3$ corresponds to Marsh Grade 1 (mild damage), $0.95 - 1.05$ to Grade 2 (moderate damage), and $< 0.95$ to Grade 3 (severe villous atrophy) (Das et al., 2019). This classification informs clinical decisions, and we assess its performance using accuracy, precision, recall, and F1 score. Lastly, to evaluate the segmentation model $S_\phi$, we use the Dice coefficient and Intersection over Union (IoU).

### 5.2 IMPLEMENTATION DETAILS

MEASURENET is built on top of the DINO-DETR with 6 encoder and 6 decoder layers, leveraging the Swin Transformer Liu et al. (2022c) as the backbone, trained with learning rate $1\times10^{-5}$. We use differentiable EMD which is based on silkhorn formulation, with $p = 2, scaling = 0.9$. Detection model ($D_\theta$) is trained with learning rate of $1 \times 10^{-4}$, batch size 8, and query dimension 4. The input images are resized to $640 \times 640$ resolution. Sinusoidal positional embeddings are used to maintain spatial information throughout the model. We update the bipartite matching in DETR Carion et al. (2020) by using Chamfer distance as the cost, allowing better matching between ground truth and predicted polyline. To enhance the model's robustness, we apply random data augmentations on the fly, including rotations of 0°, 90°, 180°, and 270°, as well as horizontal, vertical, and diagonal flips. For $L_{focal}$, $\alpha$ is set as 0.25 and $\gamma$ is set as 2. For the segmentation task, we employ SegFormer Xie et al. (2021) with the B5 backbone, training the model for 250 epochs with a batch size of 8 and a

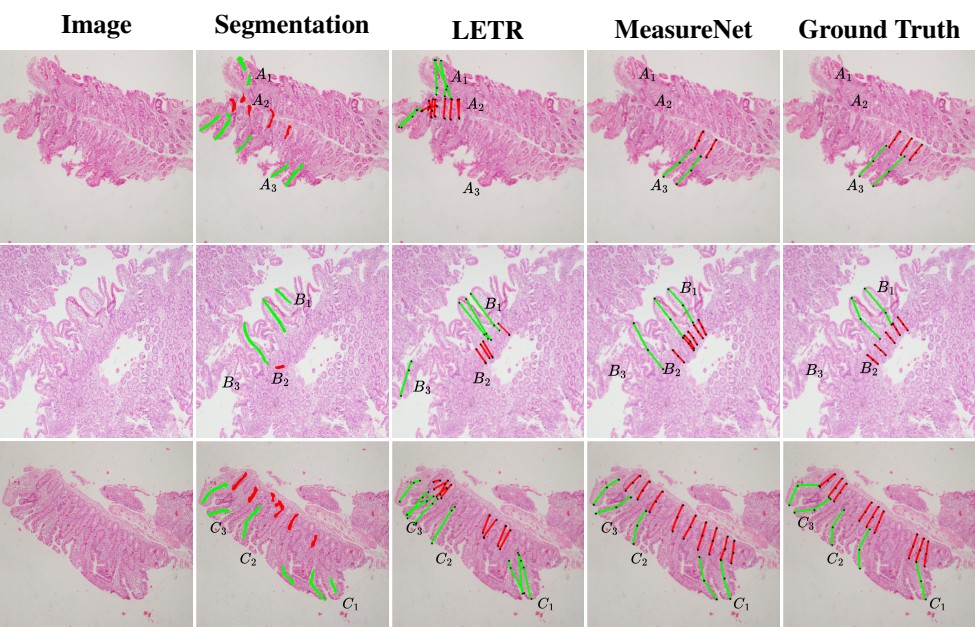

Figure 3: Qualitative performance of MeasureNet. First column shows Original Image, second, third, fourth and fifth column shows SegFormer, LETR, MeasureNet predictions and ground truth polyline respectively. Here, green and red corresponds to villi and crypts. The markers, $A_1$, $A_2$ and $B_3$ shows improvement in localization. Markers $A_3$, $B_1$, $B_2$ and $C_1$, shows improvement in partial predictions which cause error in measurement. Markers $C_2$ and $C_3$, shows better alignment and improved start-end points for villi and crypts.

learning rate of $6 \times 10^{-5}$. The final segmenter segments villi shoulder, crypt border and villi, but not crypts. The images are resized to $640 \times 640$, and we apply augmentations similar to those used in the polyline detection model, including all rotations, as well as horizontal, vertical, and diagonal flips. While training MEASURENET, we use mixup where $\lambda$ is sampled from beta distribution where $\delta$ is between [0.2, 0.4]. We did grid search on the devset for hyperparameters tuning. All models are trained on an NVIDIA A100 GPU (more details C.1).

## 6 RESULTS

Table 1: Measurement and detection results for CeDeM. Best results are mentioned in bold.

| Model | M-MAE Villi ↓ | M-MAE Crypt ↓ | MAE Ratio ↓ | Precision ↑ | Recall ↑ | mAP ↑ |
|---|---|---|---|---|---|---|
| FRNet | $68.07 \pm 5.1$ | $34.86 \pm 2.9$ | $1.11 \pm 0.12$ | $31.83 \pm 2.7$ | $27.16 \pm 3.1$ | $36.52 \pm 2.8$ |
| SegFormer | $33.05 \pm 1.75$ | $19.09 \pm 1.27$ | $0.858 \pm 0.07$ | $36.61 \pm 2.3$ | $55.72 \pm 3.6$ | $40.91 \pm 2.9$ |
| BEiT | $62.22 \pm 4.6$ | $38.82 \pm 3.8$ | $1.10 \pm 0.15$ | $37.91 \pm 2.4$ | $26.88 \pm 2.7$ | $47.71 \pm 3.2$ |
| Yolino | $62.98 \pm 4.2$ | $21.14 \pm 2.5$ | $1.65 \pm 0.20$ | $14.30 \pm 1.9$ | $54.18 \pm 3.3$ | $20.31 \pm 1.6$ |
| MapTR | $21.41 \pm 3.3$ | $19.32 \pm 1.1$ | $1.44 \pm 0.08$ | $\mathbf{54.16} \pm 0.7$ | $31.27 \pm 0.7$ | $42.93 \pm 2.5$ |
| LETR | $16.94 \pm 0.9$ | $11.81 \pm 0.6$ | $0.704 \pm 0.03$ | $39.52 \pm 2.1$ | $70.85 \pm 2.7$ | $40.39 \pm 2.4$ |
| MeasureNet | $\mathbf{14.09} \pm \mathbf{0.6}$ | $\mathbf{8.03} \pm \mathbf{0.2}$ | $\mathbf{0.406} \pm \mathbf{0.02}$ | $51.37 \pm 1.7$ | $\mathbf{79.94} \pm \mathbf{1.8}$ | $\mathbf{50.29} \pm \mathbf{1.5}$ |

**MEASURENET performs best for measurement and localization.** Table 1 compares MEASURENET with several baseline methods, including segmentation based approaches like FRNet (Ning et al., 2024), SegFormer (Xie et al., 2021) and BEiT (Bao et al., 2021). These segmentation models predict segmentation masks for villi and crypt, from which we identify contours followed by skeletonization of contours to compute final lengths. MEASURENET obtains minimum of 45 pt reduction in MAE ratio (lower is better) and 3 pt mAP improvement as compared to Segmentation based methods. Segmentation based methods rely on visual attributes of villi/crypts to make prediction, however, these attributes can be misleading in some cases where crypts are partially visible. We also compare against Yolino (Meyer et al., 2021), a lane detection model. However, we observe that anchor based approaches like Yolino struggle to capture the complex curvatures present in polylines.

Next, we adapt detection based methods MapTR (Liao et al., 2025) and LETR (Xu et al., 2021) for polyline detection. MEASURENET improves MAE ratio from 1.44 to 0.406 and mAP from changes from 42.93 to 50.29. LETR (Xu et al., 2021), originally designed for line prediction with two points is modified to predict three points instead of two. Compared to LETR, MEASURENET obtains 30 pt error reduction in MAE ratio and 10 pt mAP improvement.

Table 2: Comparison of classification metrics for binary and grade level celiac disease classification across different models. ResNet and ViT-b are baseline classifiers.

| Type | Method | Precision | Recall | F1 | Acc. |
|---|---|---|---|---|---|
| Binary | ResNet* | 36.84 | 46.67 | 41.18 | 71.01 |
| | ViT-b* | 39.12 | 49.50 | 43.78 | 76.81 |
| | SegFormer | **87.81** | 81.13 | 84.31 | 76.81 |
| | LETR | 81.03 | 87.03 | 83.92 | 73.91 |
| | MeasureNet | 82.81 | **98.14** | **89.83** | **82.66** |
| Grade | ResNet* | 28.82 | 28.87 | 28.79 | 66.67 |
| | ViT-b* | 37.22 | 36.22 | 36.06 | 72.46 |
| | SegFormer | 71.01 | 43.40 | 46.00 | 45.30 |
| | LETR | 33.75 | 34.17 | 33.33 | 70.10 |
| | MeasureNet | **83.60** | **74.32** | **75.61** | **81.42** |

MapTR and LETR struggle with measurement accuracy due to the absence of polyline based loss, measurement losses, and auxiliary masks, suggesting that global attributes like villi shoulders and crypt borders guide MEASURENET toward precise measurements. MEASURENET accurately estimates villi and crypt lengths ($103.4 \pm 23.1$ and $45.9 \pm 11.5$), closely matching ground truth ($104 \pm 25.3$ and $46.6 \pm 14.7$) (see D.6). See D.3, D.4, D.2, D.5 for ablation on number/type of points, distance metric and impact of feature aligner. We found MEASURENET statistically significantly better (p-value 0.0002 for MAE ratio) compared to the closest baseline.

**MEASURENET predictions have better curvature and villi/crypt length measurements.** Qualitatively, we illustrate MEASURENET predictions in Fig 3. We observe that MEASURENET improves the localization performance as shown by highlighted markers $A_1$, $A_2$ and $B_3$, and reduces partial predictions shown by $A_3$, $B_1$, $B_2$ and $C_1$. Partial predictions hurt model performance by affecting the measurement eventually resulting in misclassification. Finally, MEASURENET provide better alignment of predicted polyline with villi, shown by $C_2$ and $C_3$. For DeepBacs see appendix D.1.

Table 3: Measurement performance on DeepBacs.

| Method | M-MAE | Precision | Recall | mAP |
|---|---|---|---|---|
| FRNet | 25.05 | 53.42 | 97.55 | 51.21 |
| LWNet | 27.42 | 52.13 | 96.27 | 50.83 |
| SegFormer | 26.05 | 58.21 | **98.61** | 55.94 |
| LETR | 14.07 | 62.72 | 94.39 | 61.60 |
| MeasureNet | **10.19** | **65.63** | 93.81 | **64.27** |

**MEASURENET based celiac grading is accurate and interpretable.** Table 2 presents the classification performance based on the villi-to-crypt length ratio. We compare MEASURENET against standard classification approaches including ResNet50 (He et al., 2015) and ViT (Dosovitskiy et al., 2020) based models, which directly predict celiac disease categories. Unlike these direct classifiers, MEASURENET not only achieves higher accuracy but also offers interpretability, making it more suitable for clinical adoption by pathologists. Specifically, MEASURENET improves the binary classification accuracy from 73.91% to 82.66% and the multiclass (grading) accuracy from 70.10% to 81.42%. Additionally, Table 3 reports the measurement performance on the DeepBacs dataset, where MEASURENET achieves a 3.88 point M-MAE improvement over the closest baseline. Lastly, MEASURENET takes about 213 ms per image, which is quite small for an eventual use in a pathology-AI collaboration system.

Table 4: Ablation study showing impact of Polyline Loss (PL), Length Loss (LL), Part Length Loss ($PL_p$), Mask supervision (Mask), and Mask Mixup (Mix). MAE values are reported for villi, crypts, and their ratio.

| Base | $L_{EMD}$ | $L_L$ | $L_{PL}$ | Mask | Mixup | MAE-V | MAE-C | MAE-R |
|---|---|---|---|---|---|---|---|---|
| ✓ | | | | | | 20.18 | 12.62 | 0.714 |
| ✓ | ✓ | | | | | 18.33 | 11.74 | 0.574 |
| ✓ | ✓ | ✓ | | | | 17.02 | 11.13 | 0.572 |
| ✓ | ✓ | ✓ | ✓ | | | 16.43 | 11.34 | 0.563 |
| ✓ | ✓ | ✓ | ✓ | ✓ | | 19.16 | 10.23 | 0.473 |
| ✓ | ✓ | ✓ | ✓ | ✓ | ✓ | **14.09** | **8.03** | **0.406** |

## 6.1 ABLATION STUDIES

We perform ablation analysis to understand the relative contribution of different components in final performance, shown in Table 4. Starting with DINO-DETR as the baseline (row 1) and adding EM distance (row 2), we observe an improvement in MAE ratio by 0.014

pts. Introducing length loss (row 3) further reduces the MAE Villi from 18.33 to 17.02. Adding part length loss (row 4) brings MAE villi and ratio down to 16.43 and 0.572, attributed mainly to improved curvature alignment in villi polylines. Incorporating mask information (row 5) significantly improves the M-MAE for crypts, from 11.34 to 10.23, emphasizing the auxiliary mask's role in providing additional cues about villi shoulders and crypt borders.

However, reliance on the segmentation mask can lead to false positives if the mask itself contains inaccuracies. For this, we introduced mask mixup, which enhances crypt length estimation and improves robustness to false positives in the segmentation mask. Compared to row 5, MEASURENET with mask mixup achieves an error reduction of 5.07 pts in villi M-MAE, 2.2 pts in crypt M-MAE, and 0.067 pts in the ratio MAE, demonstrating gains in measurement. Table 5 highlights the role of auxiliary information. With only VS and CB (row 1), we obtain 15.4 and 8.04 MAE for villi/crypt. Adding villi (row 2) improves villi curvature improving MAE-V from $15.4 \rightarrow 14.09$. Including crypts instead (row 3) degrades MAE-C ($8.04 \rightarrow 12.07$), as crypt are only partially visible in many cases and this leads to partial segmentation, introducing noise in the detection model. Same is observed in row 4.

Table 5: Ablation study on the impact of auxiliary segmentation cues: VS (villi shoulder), CB (crypt border), V (villi), and C (crypt).

| VS-CB | V | C | MAE-V ↓ | MAE-C ↓ |
|---|---|---|---|---|
| ✓ | | | 15.4 | 8.04 |
| ✓ | ✓ | | **14.09** | **8.03** |
| ✓ | | ✓ | 15.5 | 12.07 |
| ✓ | ✓ | ✓ | 14.53 | 12.18 |

## 6.2 ERROR ANALYSIS

**Denuded villi and indistinguishable villi shoulder still a challenge.** We conduct an error analysis of MEASURENET, highlighting common errors (Fig. 4). Row 1 shows errors arising from mis-identification of the VS and CB, often due to the lack of distinct visual attributes. This results in change in ratio, and are more prominent in cases of severe CeD. Row 2 highlights errors due to false positives villi predictions. For villi measurements pathologists consider good villi with unbroken epithelial layers (outer layer). MEASURENET occasionally misclassifies denuded villi (with broken epithelial layer) as good villi. While these do not significantly affect measurements, but impact localization performance.

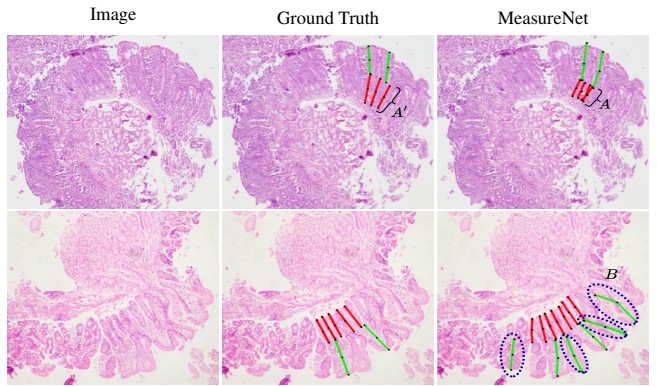

Figure 4: Two main types of errors include: 1) Missing or incorrect Villi Shoulder (VS) and Crypt Border (CB), leading to shifted crypt predictions ($A' > A$), as shown in row 1; 2) False positive predictions (B) for denuded villi, which should be excluded from measurements, as shown in row 2.

## 7 CONCLUSION

We address the challenge of identifying and grading celiac disease by accurately measuring villi and crypt length in duodenal biopsy images. Our proposed model, MEASURENET, introduces a novel polyline detection framework that combines polyline detection loss, derived from Earth Mover's distance, with measurement loss that includes both length and part-length loss components. To enhance crypt polyline detection, MEASURENET incorporates auxiliary guidance of villi shoulders and crypt borders in form of segmentation mask. Mask feature mixup prevents over-reliance on auxiliary masks, ensuring robustness and accuracy in detection. We introduce CeDeM, a novel dataset of 752 annotated duodenal biopsy images for identifying celiac disease through villi and crypt measurements. MEASURENET shows significant effectiveness on CeDeM, outperforming the closest baseline in measurement, localization, and classification metrics: improving villi-crypt length ratio error from 0.70 to 0.40, increase in mAP from 40.3 to 50.3, and in classification accuracy from 72% to 81.4%. In the future, we will develop a pathologist-assist software for increasing their productivity in assessing celiac patients. We will release both code and dataset for future research.

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

# A  DATASET DETAILS

Celiac Disease Detection and Grading through Measurement (CeDeM) has been annotated under the supervision of expert pathologists, following a consistent protocol. Due to tissue deformation and sectioning artifacts commonly introduced during histopathology slide preparation, not all regions of a slide are suitable for quantitative measurement. Histopathology slides may contain various structures such as elongated crypts, circular crypts, well formed villi with an intact epithelial layer, and denuded villi (see Fig. 5). Only regions containing elongated crypts and good villi are considered suitable for measurement. These regions are referred to as the interpretable region (highlighted by the pink bounding box in Fig. 5), and all villi and crypt annotations are performed within this region. Although the interpretable region itself is not used during model training, it is included as part of the annotation metadata to guide and standardize the annotation process.

Within these interpretable regions, villi and crypt structures are manually annotated by pathologists. For villi that are relatively straight, annotations are made using two points to mark their start and end. In cases where the villi are curved, a variable number of points, up to four, are placed along the structure, with the number of points determined by the degree of curvature to accurately capture their shape. If a villus is denuded (i.e., missing or not clearly discernible within the interpretable region), it is excluded from annotation. Crypts are annotated using two points, indicating their start and end. CeDeM is annotated by experienced pathologists and subsequently validated by senior pathologists. In instances of disagreement, one-on-one discussions are conducted to reach a consensus and finalize the annotations.

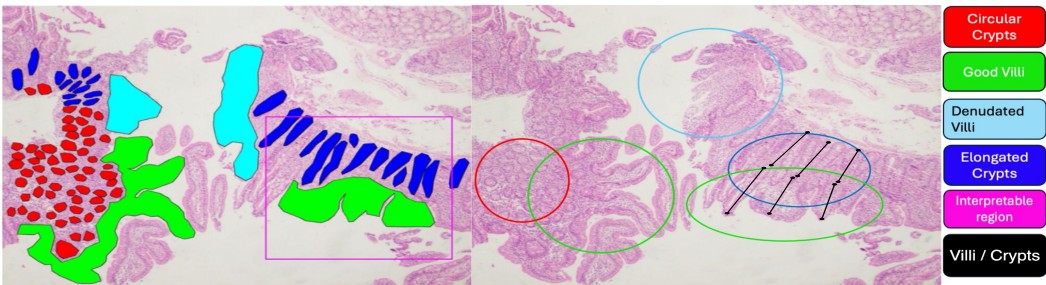

Figure 5: The left image illustrates various structures present in the histopathology slide, including circular crypts, elongated crypts, well-formed villi, and denuded villi. The region containing well-formed villi and elongated crypts is referred to as the interpretable region and is marked with a pink bounding box. The right image highlights the elongated crypts and well-formed villi that are used for measurement annotations. Note: The segmentation of elongated villi, circular villi, and other structures are shown for illustration purposes only and are not part of the dataset. The CeDeM dataset includes annotations for villi, crypts, villi shoulder, crypt border, and the interpretable region.

# B  MEASURENET

## B.1  DETECTION MODEL

DINO Zhang et al. (2022) is a DETR-based object detection model that improves training stability and accuracy through three key innovations: (1) contrastive denoising training, which adds both positive and negative noisy versions of ground truth boxes to reduce duplicate predictions; (2) mixed query selection, combining encoder-based positional queries with learnable content queries for better initialization; and (3) a look forward twice mechanism, which refines early layer parameters using gradients from later layers. DINO also adopts deformable attention for efficiency and builds on DAB-DETR Liu et al. (2022a) and DN-DETR Li et al. (2022) for query refinement and stable bipartite matching.

## B.2 SEGMENTATION MODEL

SegFormer Xie et al. (2021) is a transformer-based framework for semantic segmentation that achieves a strong balance of efficiency, accuracy, and robustness. Unlike prior methods that primarily focus on encoder design, SegFormer introduces a novel hierarchical, positional-encoding-free encoder and a lightweight all-MLP decoder. The encoder effectively produces multi-scale features and adapts to varying image resolutions without positional interpolation, while the decoder aggregates both local and global attention by combining information across layers. This design allows SegFormer to perform well across diverse benchmarks. We use SegFormer to predict Villi, Villi Shoulder (VS) and Crypt border (CB).

## B.3 MASK SUPERVISION

The segmentation model $S(\theta)$ is used to provide auxiliary information about the start and end points of villi and crypt structures. However, in some cases, we observe a mismatch between the curvature of the villi shoulder (VS) and the crypt border (CB). For instance, the villi shoulder may extend significantly beyond the crypt border, leaving no corresponding point on the CB that can be used as a valid endpoint. Such inconsistencies introduce noise into the detection model and degrade measurement accuracy.

To address this issue, we introduce a Dynamic Time Warping (DTW) loss, denoted as $L_{\text{DTW}}$, which enforces alignment between the predicted villi shoulder and crypt border contours. This loss encourages structural consistency between the two components by minimizing the cumulative alignment cost.

The DTW loss is defined as:

$$L_{\text{DTW}}(P, Q) = \min_{W} \sum_{(i,j) \in W} d(p_i, q_j),\tag{5}$$

where $P = \{p_1, p_2, \ldots, p_m\}$ and $Q = \{q_1, q_2, \ldots, q_n\}$ denote the sets of points representing the predicted villi shoulder and the crypt border contours, respectively. The term $d(p_i, q_j)$ denotes the Euclidean distance between point $p_i$ and point $q_j$, and $W$ is the optimal warping path that aligns the two sequences.

To ensure fair alignment when $\text{len}(P) \neq \text{len}(Q)$, we apply DTW over the shorter of the two sequences, using $\min(\text{len}(P), \text{len}(Q))$ to limit the alignment scope. This helps to maintain geometric consistency and improve the reliability of downstream detection tasks.

## B.4 FEATURE ALIGNER

MeasureNet employs a feature aligner module (see Fig. 6) to capture local patterns from both the image and the corresponding segmentation masks. This module is inspired by RoadFormer Huang et al. (2024) and is designed to enhance fine-grained structural information critical for accurate measurement in histopathology images.

Local structural details—such as sharp boundaries and subtle morphological variations—are essential for precise annotation and measurement, particularly for structures like villi and crypts. We use a lightweight Feature Aligner (FA) module based on the inverted residual block from MobileNetV2 Sandler et al. (2018). This module refines the image and mask features, with an emphasis on enhancing local detail without significantly increasing model complexity.

The operation of the FA module is defined as:

$$fe_I = \text{Conv}\left(\text{ReLU}\left(\text{DWConv}\left(\text{Conv}(f_I)\right)\right)\right) + f_I,\tag{6}$$

where $f_I$ denotes the input feature map, and $fe_I$ is the output with enhanced local features. After applying the FE module, we obtain feature maps for both the image and mask inputs, denoted as $fa_I$ and $fa_M$, respectively.

## B.5 FEATURE MERGER

The Feature Merger (FM) block, illustrated in Fig. 7, takes the locally enhanced image and mask features, $fa_I$ and $fa_M$, as input and produces a unified feature representation, $f_{UIM}$. First, $fe_I$

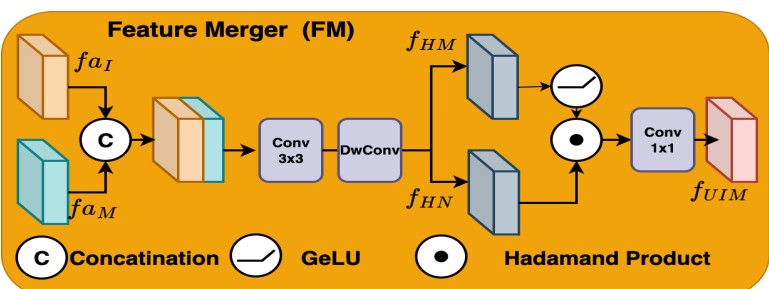

Figure 6: Feature Extraction Block

Figure 7: Feature Merger Block

and $fe_M$ are concatenated and passed through a convolutional block followed by a depthwise convolution. This results in a $4\times$ channel expansion, enriching the feature representation. The resulting tensor is then split into two parts, $f_{HM}$ and $f_{HN}$. One of these branches ($f_{HN}$) is processed through a GELU activation and then combined with the other branch ($f_{HM}$) via a Hadamard product. This interaction highlights the most relevant regions, capturing a stronger relationship between the image and mask features. To maintain consistency in dimensionality with the original input, a final $1 \times 1$ convolution is applied to restore the output channel size. The flow of features through the FM block is detailed in Equations 7–12.

$$f_C = \text{Concat}(f_I, f_M) \tag{7}$$

$$f_1 = \text{Conv}_{3\times3}(f_C) \tag{8}$$

$$f_2 = \text{DWConv}_{3\times3}(f_1) \tag{9}$$

$$H_m, H_n = \text{Split}(f_2) \tag{10}$$

$$f_3 = H_m \cdot \text{GELU}(H_n) \tag{11}$$

$$f_{\text{out}} = \text{Conv}_{1\times1}(f_3) \tag{12}$$

## C    EXPERIMENTAL SETTING

### C.1    IMPLEMENTATION DETAILS

Yolino, originally designed for single-class lane predictions, was adapted for our task by training two separate models: one for villi and another for crypts. The polyline annotations were reformatted to align with lane detection requirements, and each image was divided into a $32 \times 32$ grid to ensure that the start and end coordinates of the polyline fell within the grid. Yolino was trained for 80

epochs using the Adam optimizer with a learning rate of $1 \times 10^{-4}$. We trained BEiT using the *beit-base-finetuned-ade-640-640* backbone for 200 epochs with a learning rate of $1 \times 10^{-6}$ and a batch size of 4. For LETR, we employed DINO-DETR as the base architecture with a Swin Transformer backbone. LETR was trained for 150 epochs with a batch size of 2, a learning rate of $1 \times 10^{-4}$, a query dimension of 4, and number of queries being 900. We tried pre-training LETR however we did not see any improvement in the performance. LETR is trained with same augmentations as MeasureNet.

For DeepBacs, LETR, and MeasureNet, training is conducted for 50 epochs, with convergence observed around 30 epochs, utilizing data augmentations. Segmentation models, including SegFormer, BEiT, and FRNet, are also trained for 50 epochs with learning rates of $1 \times 10^{-6}$, $6 \times 10^{-5}$, and $1 \times 10^{-4}$, respectively.

## C.2 EVALUATION METRIC

To evaluate the quality of length measurements, we compute the Matched Mean Absolute Error (M-MAE). M-MAE accounts for both over-prediction and under-prediction by incorporating bidirectional matching between predicted and ground truth polylines. It is defined as:

$$\text{M-MAE} = \max(\text{MAE}_{G \to P}, \text{MAE}_{P \to G})$$

Here, $\text{MAE}_{G \to P}$ denotes the mean absolute error computed by matching each ground truth polyline to its closest predicted polyline and measuring the absolute difference in their lengths. Conversely, $\text{MAE}_{P \to G}$ is calculated by matching each predicted polyline to its closest ground truth polyline.

In cases of false positives—where a predicted polyline does not correspond to any ground truth—the error contributes to $\text{MAE}_{P \to G}$, as the length difference with the nearest ground truth increases. In the case of false negatives, $\text{MAE}_{G \to P}$ increases, as the ground truth polyline matches with the next closest predicted polyline. Notably, when multiple predicted polylines align closely with a single ground truth polyline, M-MAE remains largely unaffected, as only the best match is considered in each direction.

# D RESULTS

## D.1 QUALITATIVE PERFORMANCE

Fig. 9 and Fig. 10 present qualitative comparisons of MeasureNet with other baseline methods on the CeDeM and DeepBacs datasets, respectively. MeasureNet demonstrates consistent improvements in both localization and measurement accuracy. The segmentation-based approach using SegFormer suffers from discontinuous predictions, which negatively affect the measurement of structures. LETR performs better but often produces misaligned predictions—such as slanted polylines that do not follow the true object boundaries, leading to measurement errors. In contrast, MeasureNet provides more continuous and well-aligned predictions, with better curvature, resulting in more accurate measurements.

## D.2 EFFECT OF TYPE OF POINT SELECTION

We train MeasureNet using start–middle–end point annotations. In cases where only two points are available, the middle point is computed as the midpoint between the start and end points. For instances with more than three annotated points, we sample one representative point between the start and end to maintain structural consistency. Alternatively, in the absence of a computed middle point, the first or last point can be replicated to maintain the expected input format.

Table 6 compares both strategies. Our results show that including a computed middle point improves the curvature representation for villi, enabling a more accurate depiction of their natural morphology. For crypts, while adding a middle point may occasionally introduce slight curvature or bending, the effect is minimal. Overall, the benefits of improved structural representation outweigh these minor artifacts (see Fig. 10, row 2 - row3, column 4).

| Model | MAE Villi ↓ | MAE Crypt ↓ | MAE Ratio ↓ |
|---|---|---|---|
| Start-End | 16.58 | **8.01** | 0.4722 |
| Middle Point | **14.09** | 8.03 | **0.4060** |

Table 6: Comparison of measurement results based on different methods for selecting the third point in polylines when the original annotation contains only two points. Middle Point refers to the midpoint of the line segment connecting the two points, while Start-End Point involves duplicating either the start or end point.

### D.3 EFFECT OF DISTANCE METRIC

We evaluate the impact of different distance metrics Chamfer Distance, Manhatten Distance and Earth Mover's Distance (EMD) on the performance of MeasureNet. Results are presented in Table 7. Earth Mover's Distance, also referred to as Wasserstein Distance, quantifies the minimal cost required to transform one distribution into another, taking into account both spatial displacement and distributional mass.

Our findings indicate that using EMD leads to improved measurement accuracy compared to Chamfer Distance, particularly in capturing the finer structural details of villi and crypts. This suggests that EMD provides a more discriminative alignment signal during training, resulting in more precise predictions.

Table 7: Comparison of measurement errors using different distance metrics.

| Model | MAE Villi ↓ | MAE Crypt ↓ | MAE Ratio ↓ |
|---|---|---|---|
| Chamfer | 14.17 | 8.22 | 0.4682 |
| Manhattan | 14.98 | 8.04 | 0.4092 |
| Earth Mover | **14.09** | **8.03** | **0.4060** |

### D.4 EFFECT OF NUMBER OF POINTS SELECTION

We analyzed the impact of varying the number of points used to represent a polyline for villi and crypt structures using our best performing MEASURENET. Table 8 shows the measurement errors for different configurations. During training, MEASURENET we used a 3 point polyline representation. We evaluated MEASURENET trained with 2, 3, and 4 points against the original annotations. The results reveal an improvement of 8.9 pts in measurement accuracy when increasing from a 2 point to a 3 point polyline. However, moving from a 3 point to a 4 point polyline showed negligible additional benefit. This indicates that a 3 point polyline is sufficient for capturing the villi-crypt structure.

Table 8: Effect of number of points for polyline.

| Points | M-MAE Villi | M-MAE Crypt | MAE Ratio |
|---|---|---|---|
| 2 | 23.06 | 10.51 | 0.676 |
| 3 | **14.09** | 8.03 | **0.406** |
| 4 | 24.71 | **7.65** | 0.665 |

### D.5 BENEFIT OF FEATURE ALIGNER (FA) MODULE

Table 9 presents a comparison between MEASURENET and MEASURENET without the Feature Aligner (FA) block. Since the backbone (Swin Transformer) is pretrained on natural images, the extracted features may not fully capture the fine-grained structural details of villi and crypt regions. Introducing the FA block before the feature merger refines these backbone features from both images and masks, leading to much better representations.

Table 9: Comparison of measurement errors using different distance metrics.

| Model | MAE Villi ↓ | MAE Crypt ↓ | MAE Ratio ↓ |
|---|---|---|---|
| MeasureNet | **14.09** | **8.03** | **0.406** |
| MeasureNet w/o FA | 15.43 | 8.72 | 0.420 |

## D.6 LENGTH PREDICTIONS

Figure 8(a) shows the distribution of villi and crypt lengths for both ground truth and MEASURENET predictions. The histograms and boxplots indicate that MEASURENET closely follows the ground truth distribution across both short and long lengths. Figure 8(b) summarizes the mean and standard deviation, showing that the predicted lengths for villi (103.4±23.1) and crypt (45.9±11.5) are highly consistent with the ground truth values (104.4±25.3 and 46.6±14.7, respectively).

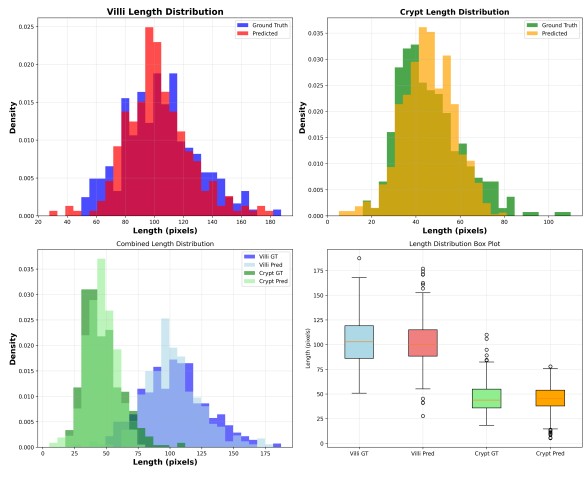

|  | Villi | Crypt |
|---|---|---|
| GT | 104.4±25.3 | 46.6±14.7 |
| Pred | 103.4±23.1 | 45.9±11.5 |

(a) Length analysis plot.

(b) Comparison of Villi and Crypt length for ground truth and predictions

Figure 8: Length predictions: (a) visualization of length analysis, (b) comparison of Villi and Crypt ground truth vs. predictions.

## D.7 COMPUTATIONAL COST

Table 10 compares the inference time and accuracy of MEASURENET with and without the segmentation branch. Removing the segmentation branch reduces inference time from 213 ms to 134 ms per image but results in higher errors for villi, crypt, and ratio predictions. Since inference times are relatively small in both cases, the improved length predictions is more beneficial for pathologists.

Table 10: Inference time computational cost between MeasureNet without and with segmentation branch

| Model | Time(msec) | MAE Villi ↓ | MAE Crypt ↓ | MAE Ratio ↓ |
|---|---|---|---|---|
| MeasureNet w/o seg branch | 134 | 16.43 | 11.34 | 0.563 |
| MeasureNet | 213 | 14.09 | 8.03 | 0.406 |

## D.8 QUALITATIVE PERFORMANCE

Figures 9 and 10 present a qualitative comparison of different models with MEASURENET. Segmentation-based method (segformer) produce discontinuous predictions, which negatively affect length measurements, while LETR often yields misaligned length predictions. In contrast, MEASURENET generates accurate and well-aligned predictions. D

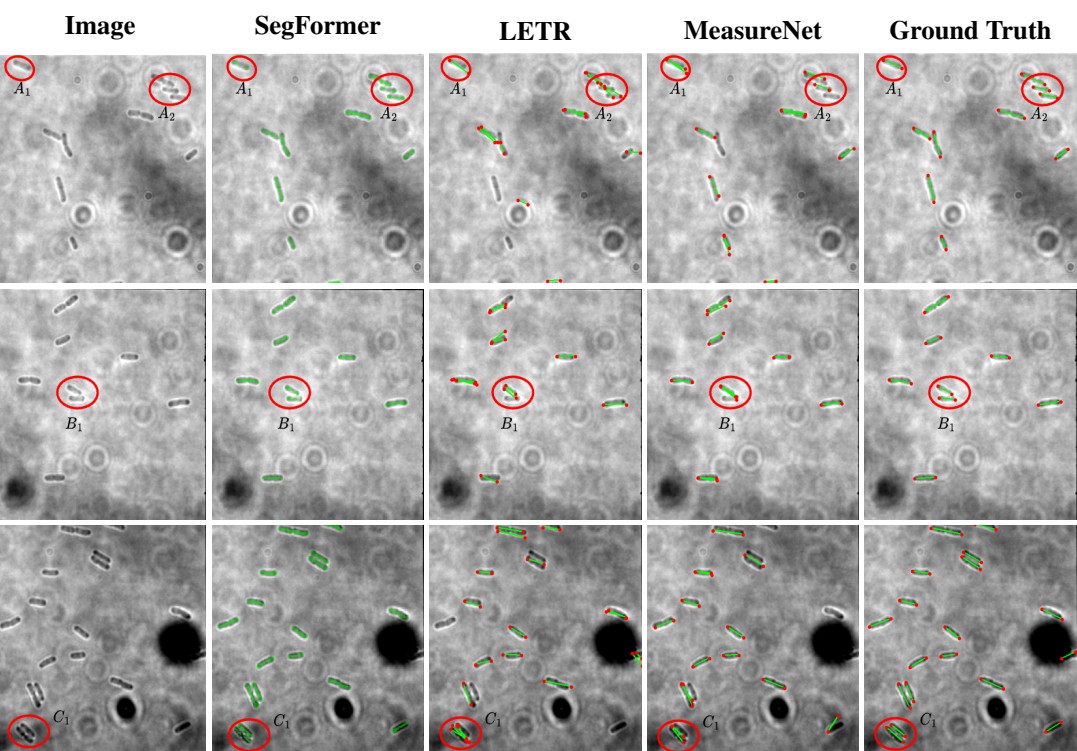

Figure 9: Qualitative comparison of model performance for bacterial length measurement. The first column shows the original image, while the second, third, fourth, and fifth columns show predictions from SegFormer, LETR, MeasureNet, and the ground truth polyline, respectively. Markers $A_1$, $A_2$, $B_1$, and $C_1$ highlight regions where measurement accuracy improves across models. SegFormer, a segmentation-based method, produces discontinuous predictions, which negatively affect the estimated bacterial length. LETR performs better but still generates predictions that are misaligned with the actual bacterial structures, as indicated by marker $A_2$. MeasureNet demonstrates better performance in terms of both measurement accuracy and spatial localization.

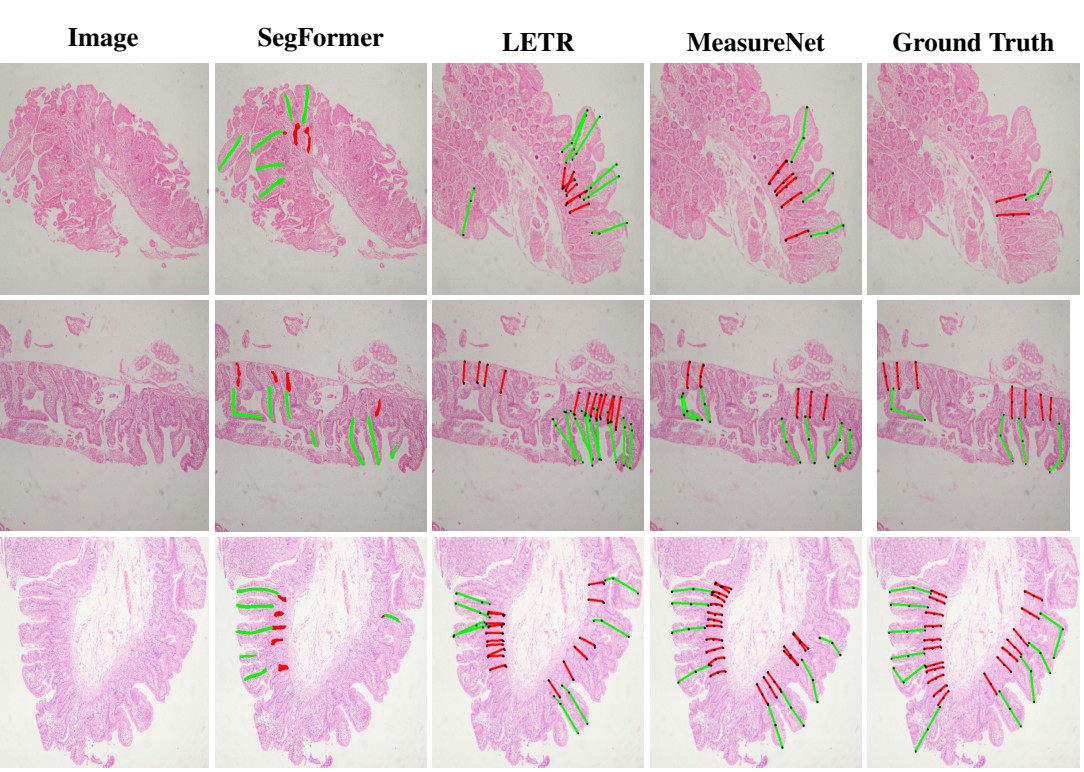

Figure 10: Qualitative performance comparison of various models for villi and crypt detection and localization. Columns show the original image, SegFormer predictions, LETR predictions, MeasureNet predictions, and ground truth, respectively. Green and red represent villi and crypt annotations.

