# OpenReview forum: "MeasureNet: Polyline Detection Based Measurement for Celiac Disease Identification and Grading"
_ICLR.cc/2026/Conference — Submitted to ICLR 2026_

### Official Review · Reviewer_LHoP · 2025-10-24

**Soundness:** 2
**Presentation:** 2
**Contribution:** 2
**Rating:** 2
**Confidence:** 4

**Summary:**

The paper introduces MeasureNet, a novel polyline detection framework designed to measure villi and crypt lengths directly from biopsy images without relying on full segmentation.

**Strengths:**

This paper proposes an interesting task that has research significance in Diagnosing celiac disease (CeD).

**Weaknesses:**

There are lots of unclear details regarding to the training process and problems in presentations

**Questions:**

1.The task definition is unclear. The work seems to target a structured detection problem rather than conventional object detection. It would help to add a short subsection in Section 3 that clearly defines the task, input/output format, and evaluation target.
2.The training process is not well explained. Are the strong and weak segmenters trained jointly or separately? During detector training, are auxiliary masks provided as input? The overall pipeline should be clarified.
3.In Figure 3, the “Segmentation” block should be labeled as SegFormer for consistency.
4.The paper does not explain how different loss terms are balanced. Please describe or justify the weighting strategy.
5.The dataset is highly imbalanced. Did the authors consider any resampling or weighting strategy to address this issue?
6.It would be useful to include a comparison of training cost or efficiency with baseline models.

---

> ### Author Response · Authors · 2025-12-02
>
> > Q1: The task definition is unclear. The work seems to target a structured detection problem rather than conventional object detection. It would help to add a short subsection in Section 3 that clearly defines the task, input/output format, and evaluation target.
>
> We would like to point the reviewer to paragraph 1 for Section 3 which clarifies the task, input/output and target. Line 149-153 reads as follows:
>
> Based on this clinical insight, we formulate our technical problem as follows: given a biopsy image \( I \), output a set of polylines $\( P = \{P_1, P_2, \dots, P_W\} \)$ with corresponding class labels $\( C = \{c_1, c_2, \dots, c_n\} \)$ and their pixel lengths, for two classes -- villi and crypts. The training data consists of polyline annotation for (approximate) bisectors of well-oriented villi and crypts, and also segmentation mask annotation for Villi Shoulder and Crypt Border.
>
> Additionally, we evaluate against the ground truth polylines on three aspects 1) Localization, 2) Measurement and 3) Celiac classification (mentioned in section 5.1 ).
>
> > Q2 : The training process is not well explained. Are the strong and weak segmenters trained jointly or separately? During detector training, are auxiliary masks provided as input? The overall pipeline should be clarified.
>
> Kindly refer to paragraph 3 of Methods, which defines the training strategy of MeasureNet (specifically line 164 where the training of segmentation models is described). For context, paragraph 3 reads as:
>
> ​​“The detector module of MeasureNet is based on the DINO-DETR method, with additional loss functions beyond standard localization and classification losses. These include earth mover's distance for comparing curvature and measurement losses for comparing lengths of the predicted polylines with the ground truth. The segmentor is trained **separately** using auxiliary annotation, optimized with Dice and cross-entropy losses, and additionally a Dynamic Time Warping (DTW) loss for better alignment between corresponding points on the villi shoulder and the crypt border. MeasureNet trains the modules component-wise, first training the segmentor and then the detector. To reduce exposure bias — that is, over-reliance on auxiliary segmentation masks — MeasureNet applies feature mixup between mask features obtained from a weak and a strong segmenter.”
>
> > Q3: In Figure 3, the “Segmentation” block should be labeled as SegFormer for consistency.
>
> We have updated the caption of the figure which now reads as “During training, given an input image (I), the segmentation models (SegFormer)”
>
>
> > Q4: The paper does not explain how different loss terms are balanced. Please describe or justify the weighting strategy.
>
> “We simply add all terms, and do not introduce any hyperparameters in the final loss, to save on hyperparameter tuning.”
> We went back and did the hyper-parameter search,
> Let us assume that $L = \lambda_{1} * L_{emd} + \lambda_{2} * L_{L} + \lambda_{3} * L_{PL}$. Overall the existing weighting method works best in our case.
>
> | $\\lambda\_{1}$ | $\\lambda\_{2}$ |$ \\lambda\_{3}$ | MAE Villi  | MAE Crypt |
> | :---- | :---- | :---- | :---- | :---- |
> | 1 | 1 | 1 | **14.09** | 8.09 |
> | 2 | 1 | 1 | 16.09 | 8.08 |
> | 1 | 2 | 1 | 16.03 | 8.82 |
> | 1 | 1 | 2 | 15.50 | 8.30 |
> | 1 | 2 | 2 | 16.78 | **7.90** |
>
> > Q5: The dataset is highly imbalanced. Did the authors consider any resampling or weighting strategy to address this issue?
>
> CeDeM contains 4211 crypts and 2617 villi. In order to mitigate this class imbalance issue we use Focal Loss (line 250-253 ) which states that: “Focal Loss helps mitigate the effect of class imbalance by down-weighting the loss assigned to well-classified examples, focusing more on hard to classify instances.”
> For Focal loss, α is set as 0.25 and γ is set as 2.
>
> > Q6 : It would be useful to include a comparison of training cost or efficiency with baseline models.
>
> We carefully adapted all baseline models for the polyline detection task. Baselines such as Yolino, LETR, and MapTR require approximately 6–8 hours of training, whereas MeasureNet takes around 8–10 hours. This additional cost stems from using both strong and weak segmentation models during training and training Image-mask feature integrator block; however, during inference only the strong segmentor is used.
>
> Furthermore, the table below presents inference times for SegFormer, LETR, and MeasureNet, illustrating that the modest additional overhead results in significantly improved measurement reliability,  which is especially important in medical diagnosis
>
> | Model  | Time (msec) | MAE Villi  | MAE Crypt | MAE Ratio |
> | :---- | :---- | :---- | :---- | :---- |
> | SegFormer  | 65.81 | 33.05 | 19.09 | 0.858 |
> | LETR | 130 | 16.94 | 11.81 | 0.704 |
> | MeasureNet | 213 | 14.09 | 8.09 | 0.406 |
> All results are computed on A100 GPU.

---

### Official Review · Reviewer_S26Q · 2025-10-29

**Soundness:** 3
**Presentation:** 3
**Contribution:** 2
**Rating:** 2
**Confidence:** 5

**Summary:**

This paper presents MeasureNet, a polyline detection framework that integrates polyline localization with object-driven losses for measurement tasks. It further employs auxiliary segmentation guidance and a mask feature mixup to handle partially visible crypts and reduce noise dependence.

**Strengths:**

The paper is clearly written, with a precise formulation of the problem and convincing demonstrations.

**Weaknesses:**

1. The motivation for using both a strong and weak segmentor is not clearly explained.
2. The introduction of two segmentation models inevitably increases computational overhead — the paper lacks analysis of the training and inference efficiency.
3. The method presents limited technical novelty. The combination of the strong/weak segmentors and the mixup strategy appears conceptually similar to a form of data augmentation.
4. The proposed framework is built upon DINO-DETR, a strong pre-trained model, whereas the competing baselines in Tables 1 and 2 do not seem to use such a pretrained backbone, raising concerns about fairness in comparison. It remains unclear whether the observed performance gains mainly stem from DINO-DETR itself.

**Questions:**

Will CeDeM be made publicly available?

---

> ### Author Response · Authors · 2025-12-02
>
> > W1:The motivation for using both a strong and weak segmentor is not clearly explained.
>
> Thank you for highlighting this point. We would like to bring reviewers attention to line 71-78 in Introduction Section. In summary,  crypts are frequently faint, partially occluded, or visually ambiguous, making them difficult to detect directly from raw tissue appearance. Therefore (like pathologists), we use segmentation masks of villi shoulder and crypt border as auxiliary guidance which essentially provides the model where crypts begin and end.
>
> The motivation for using both strong and weak segmentors is to prevent the detector from becoming over-dependent on perfect segmentation. If the model only sees strong (high-quality) masks during training, it will fail when real-world segmentation is imperfect. By introducing a weak segmentor and performing feature-level mixup between strong and weak masks, we intentionally expose the detector to segmentation errors. This reduces exposure bias and improves robustness of crypt length estimation under varying segmentation quality. To the best of our understanding, we are the first to use mixup between strong and weak segmentation masks for reducing exposure bias in training vision models.
>
> > W2: The introduction of two segmentation models inevitably increases computational overhead — the paper lacks analysis of the training and inference efficiency.
>
> We have addressed the computational overhead in Section D7 (Table 10), supplementary lines 954–964. To reiterate here: during training MeasureNet uses both strong and weak segmentation models, while during inference only the strong segmentor is used. The detector alone runs at 134 ms per image, and with segmentation guidance it runs at 213 ms which is still well below one second and fully practical for clinical workflow with negligible waiting time for pathologists. For clarity, putting Table 10 here, which demonstrates that the modest additional overhead yields significantly improved measurement reliability.
>
> | Model  | Time (msec) | MAE Villi  | MAE Crypt | MAE Ratio |
> | :---- | :---- | :---- | :---- | :---- |
> | MeasureNet w/o segmentation head | 134 | 16.43 | 11.34 | 0.56 |
> | MeasureNet | 213 | 14.09 | 8.09 | 0.406 |
>
> For Training time: We train segmentation models for 50 and 250 epochs, taking close to ~1.5 and ~8 hr respectively. For training MeasureNet, we 150 epochs which takes ~12-15 hrs on A100 GPU.
>
> > W3: The method presents limited technical novelty. The combination of the strong/weak segmentors and the mixup strategy appears conceptually similar to a form of data augmentation.
>
> As stated in W1 for reviewer TYqi : The novelty lies in both the task formulation and the methodological contributions. We introduce a new task of polyline-based structural length measurement for Celiac Disease (CeD) grading, supported by a novel expert-annotated dataset (CeDeM) with villi/crypt polylines and auxiliary VS/CB annotations. Methodologically, we propose MeasureNet, a polyline centric detection framework with measurement driven losses (EMD, length, and part-length) explicitly designed for accurate measurement. We further leverage pathology driven VS/CB cues to guide crypt estimation, and to the best of our knowledge we are the first to mitigate segmentation exposure bias via feature-level weak and strong mixup.
> Moreover, we are the first to enable measurement based CeD grading, improving the current diagnostic paradigm. While developed for celiac disease, the approach generalizes to other medical imaging domains where accurate morphological length assessment is required.
>
> > W4: The proposed framework is built upon DINO-DETR, a strong pre-trained model, whereas the competing baselines in Tables 1 and 2 do not seem to use such a pretrained backbone, raising concerns about fairness in comparison. It remains unclear whether the observed performance gains mainly stem from DINO-DETR itself.
>
> As detailed in Supplementary:  Implementation Details (lines 807–820), we carefully adapted each baseline to our task: for Yolino, we trained separate models for villi and crypts, reformatted polylines to match lane-detection format, and used a 32×32 grid to ensure proper endpoint alignment. For LETR and MapTR, we used the same DINO-DETR + Swin backbone as MeasureNet and matched the output configuration to predict 3-point polylines, ensuring architectural parity. These modifications were applied across both datasets (CeDeM and DeepBacs), ensuring that all baselines were given every opportunity to perform well within the task constraints.
>
> > Q1: Will CeDeM be made publicly available?
>
> Yes, CeDeM along the MeasureNet will be publically available to facilitate the further research in Celiac disease detection and grading.

---

### Official Review · Reviewer_uYFQ · 2025-10-31

**Soundness:** 4
**Presentation:** 2
**Contribution:** 3
**Rating:** 6
**Confidence:** 3

**Summary:**

This work proposes a new method for polyline regression using MeasureNET. They showed that using segmentation information from the villus and crypt regions helps the detector to identify and measure the lengths of the polylines. By analyzing the ratio between villus and crypt, they could perform celiac disease grading with results that outperform the state-of-the-art, even when compared with classification models.

**Strengths:**

- The authors collected a new dataset of histological images with segmentation annotation of the villi/crypts for the study of celiac disease. The dataset will be made publicly available upon possible acceptance.

- A cascade of segmentation and object detection frameworks outperforming state-of-the-art polyline detection is proposed.

-  The earth mover's distance was applied to complement the most common losses used for object detection, showing that this term improves polyline detection performance.

- Ablation studies were performed to assess the value of each contribution to the work, showing that the introduced methods, in fact, improve performance.

**Weaknesses:**

- Presentation can be improved regarding text syntax and placement of figures and tables. Tables and figures are often within the text, which compromises the paper's readability.
- The paper's main strength lies in combining segmentation information to improve detection. Nevertheless, this might have limited application.
- Some of the results can be better discussed to solidify the contributions.

**Questions:**

- In Section 3, the authors said: "measurement losses to compare lengths of the predicted polylines with the gold". What does gold mean? Gold-standard? This term is repeated along the manuscript.

- The authors mentioned that the weak segmentation model is trained with 50% of capacity. Can the authors give more information on that? It would also be valuable to present the metrics for the weak and strong segmentation models.

- In Table 1, it is shown that MeasureNET outperforms the state-of-the-art. Nevertheless, the proposed method uses the segmentation information to leverage its detection capabilities, which the other methods do not benefit from. Can the features outputted by the feature merger (fUIM) be incorporated into lane detection models such as LETR? If so, a small evaluation on the performance of LETR would be
valuable to assess if reducing the exposure bias also improves other models.

- When comparing with state-of-the-art lane detectors, are the other detection models fine-tuned using the earth mover&#39;s distance in the loss function? Especially for DeepBacs, where no segmentation is needed, it would be valuable to see the impact of this proposed term on the other models.

- In the related works, it is said that the primary tool for diagnosing CeD is the IEL counting, and a model (DeGPR) performs classification based on this counting. Why are the results for DeGPR not present in the experiments section? Since measuring the villi/crypt is just an alternative approach, a comparison between a model based on the standard procedure would be valuable for the work.

---

> ### Author Response · Authors · 2025-12-02
>
> > W1: Presentation can be improved regarding text syntax and placement of figures and tables. Tables and figures are often within the text, which compromises the paper's readability.
>
> Thanks for your feedback, we understand the table/figures within text might impact the readability, however given the page limit, we wanted to put main results in the main papers as reviewers sometimes overlook the supplementary material. If there is a specific suggestion then kindly let us know we will try to update the manuscript.
>
> > W2: The paper's main strength lies in combining segmentation information to improve detection. Nevertheless, this might have limited application.
>
> We agree that incorporating segmentation information is application specific, but this is precisely what makes it unique. MeasureNet demonstrates how global context (inspired from pathology insight) such as villi shoulder and crypt border, can be meaningfully integrated into the detection framework to improve villi /crypt length detection and measurement.
> This approach can generalize to cases where global information can improve the local predictions.
>
> > W3: Some of the results can be better discussed to solidify the contributions.
>
> If you can point out which particular case you are talking about, we will look into it and  update the manuscript if required.
>
> > Q1: In Section 3, the authors said: "measurement losses to compare lengths of the predicted polylines with the gold". What does gold mean? Gold-standard? This term is repeated along the manuscript.
>
> Yes, gold referred to the gold-standard annotations, i.e., the ground-truth polylines. To avoid ambiguity, we have replaced all occurrences of “gold” with “ground truth” in the manuscript (lines 164, 232, and 233), with changes marked in red.
>
> > Q2: The authors mentioned that the weak segmentation model is trained with 50% of capacity. Can the authors give more information on that? It would also be valuable to present the metrics for the weak and strong segmentation models.
>
> Here, strong and weak segments indicate the level of training. We train a strong segmentation model for 250 epochs, which saturates around 100 epochs. In order to train a weak segmentation model we train it for 50 epochs. The weak model is trained to a level that it understands the segmentation tasks but can make errors which we may see at the test time that mitigates the exposure bias issue. The terms of metrics following the table  provide the dice coefficient for both the models.
>
> | Segmentation Model | Dice  |
> | :---- | :---- |
> | Weak Segmentation $S\_{w}$ | 32.33 |
> | Strong Segmentation $S\_{s}$ | 40.45 |
>
>
> > Q3: In Table 1, it is shown that MeasureNET outperforms the state-of-the-art. Nevertheless, the proposed method uses the segmentation information to leverage its detection capabilities, which the other methods do not benefit from. Can the features outputted by the feature merger (fUIM) be incorporated into lane detection models such as LETR? If so, a small evaluation on the performance of LETR would be valuable to assess if reducing the exposure bias also improves other models.
>
> We modified LETR to incorporate segmentation mask based features coming from the feature merger block. We can see from the table below that with the segmentation mask, there is improvement in crypts and villi detection. However, a gap still exists as Measurement specific losses are missing.
>
> | Method  | MAE Villi | MAE Crypt | MAE Ratio | Precision  | Recall | mAP |
> | :---- | :---- | :---- | :---- | :---- | :---- | :---- |
> | LETR | 16.94 | 11.81 | 0.704 | 39.52 | 70.85 | 40.39 |
> | LETR \- Seg | 15.34 | 9.16 | 0.653 | 46.72 | 84.15 | 44.71 |
> | MeasureNet | 14.09 | 8.03 | 0.406 | 51.37 | 79.74 | 50.29 |
>
>
> > Q4 : When comparing with state-of-the-art lane detectors, are the other detection models fine-tuned using the earth mover's distance in the loss function? Especially for DeepBacs, where no segmentation is needed, it would be valuable to see the impact of this proposed term on the other models.
>
> When training for CeDeM (Celiac Disease dataset introduced by us) or Deepbacs, baseline models are modified to predict the same polyline structure as MeasureNet and wherever possible, use that same backbone. Measurement driven losses like $L\_{emd}$ are our contributions hence baseline are not training with these losses. We conducted the experiment of deepbacs (as asked), and observed that adding $L\_{emd}$ improves the MAE from 14.09 → 13.90 (\~1pt improvement). Hence these losses are general and can be used with other models as well.
>
> | Method | MAE | Precision  | Recall | mAP |
> | :---- | :---- | :---- | :---- | :---- |
> | LETR | 14.09 | 62.72 | 94.61 | 61.60 |
> | LETR \+ $L\_{emd}$ | 13.90 | 63.49 | 93.58 | 62.47 |
> | MeasureNet | 10.19 | 65.63 | 93.81 | 64.27 |

---

> > ### Author Response · Authors · 2025-12-02
> >
> > > Q5: In the related works, it is said that the primary tool for diagnosing CeD is the IEL counting, and a model (DeGPR) performs classification based on this counting. Why are the results for DeGPR not present in the experiments section? Since measuring the villi/crypt is just an alternative approach, a comparison between a model based on the standard procedure would be valuable for the work.
> >
> > Thanks for pointing this out. We would like to clarify that although both DeGPR and MeasureNet operate on duodenum biopsy images, however they do so at different microscopic resolutions—20× and 4× respectively. DeGPR is applied at 20× magnification and bases its prediction on intraepithelial lymphocyte (IEL) counting, whereas MeasureNet operates at 4× magnification and performs Celiac Disease (CeD) grading based on villi–crypt measurement. Due to this their results are not mentioned in the paper.
> >
> > In standard pathology workflow, increased IEL count is used to indicate the presence of CeD,whereas villi/crypt length measurement reflects the severity of the CeD.

---

### Official Review · Reviewer_TYqi · 2025-10-31

**Soundness:** 3
**Presentation:** 2
**Contribution:** 2
**Rating:** 4
**Confidence:** 3

**Summary:**

This paper addresses a specific problem in medical image analysis, villus and crypt detection. The key objective is not to achieve full segmentation but rather, guided by pathologists’ insight that precise segmentation of villi and crypts is not essential in clinical practice. The proposed architecture, MeasureNet, is constructed from several established components, incorporating existing network architectures or elements from such network architectures (e.g. SegFormer, DINO, inverted residual block from  MobileNetV2). In addition, the paper introduces a novel dataset, CeDeM, along with corresponding annotations. The performance of the proposed method is evaluated both on this dataset and on a second public dataset addressing a different task.

**Strengths:**

The insight to simplify the task to polyline detection makes this otherwise complex medical image analysis problem far more tractable for practical applications. The paper presents an interesting and well-motivated solution to the reduced task.

The use of mixup between strong and weak segmentation masks to mitigate exposure bias during model training appears to be a novel and promising approach.

Furthermore, the introduction of the CeDeM dataset, along with its corresponding annotations, represents a valuable contribution to the research community.

**Weaknesses:**

The methodological contribution of this work is relatively incremental. The proposed MeasureNet architecture is assembled from several established components, combined with various adaptations of the loss function.

While the selected components appear reasonable, the rationale behind their specific choice is not always clearly articulated.

A considerable number of additional terms are introduced into the loss functions. Although these terms aim to capture different aspects of the task, their relative weighting becomes increasingly problematic. For instance, the total segmentation loss is defined as Lseg = Ldice +Lce +LDTW, where the last term operates on a significantly different numerical scale. Without appropriate normalization, simply assigning all weighting factors to one may render some terms ineffective. A similar issue arises in the final training loss for Dθ. The authors state: “We simply add all terms, and do not introduce any hyperparameters in the final loss, to save on hyperparameter tuning”. However, this simplification may inadvertently diminish the impact of certain loss components.

The comparison with other methods, referred to as baselines, is also somewhat misleading. The compared approaches are “general-purpose” methods that have not been tailored to this specific task. Consequently, the reported superior performance of the proposed model is not entirely unexpected.

**Questions:**

The Earth Mover’s Distance (EMD) exhibits issues with smoothness and differentiability. Should the authors consider using a differentiable alternative, such as the Sinkhorn distance (entropic-regularized EMD)?

How critical are the many loss terms used in training? Which terms contribute most to performance, and does tuning their weights substantially affect results?

What is the LWNet reported in Table 3? This model is not described in the paper.

Table 4 suggests that mask augmentation and mixup drive most of the performance gains; the contribution of the three L terms is unclear. Are these loss components truly essential?

---

> ### Author Response · Authors · 2025-12-02
>
> > W1: The methodological contribution of this work is relatively incremental. The proposed MeasureNet architecture is assembled from several established components, combined with various adaptations of the loss function.
>
> The novelty lies in both the task formulation and the methodological contributions. We introduce a new task of polyline-based structural length measurement for Celiac Disease (CeD) grading, supported by a novel expert-annotated dataset (CeDeM) with villi/crypt polylines and auxiliary Villi Shoulder (VS)/Crypt Border (CB) annotations. Methodologically, we propose MeasureNet, a polyline centric detection framework with measurement driven losses (EMD, length, and part-length) explicitly designed for measurement. We further leverage pathology driven VS/CB cues to guide crypt estimation, and to the best of our knowledge we are the first to mitigate exposure bias via feature level weak and strong mixup. Moreover, we are the first to enable measurement based CeD grading, improving the current diagnostic paradigm. While developed for celiac disease, the approach generalizes to other medical imaging domains where accurate length assessment is required.
>
> > W2: While the selected components appear reasonable, the rationale behind their specific choice is not always clearly articulated.
>
> The choice of each component was driven by specific characteristics of the task and validated by ablation studies. DINO-DETR provides robust polyline localization due to its object detection capability; the measurement-driven loss terms (EMD, length, and part-length) were introduced to explicitly capture villi/crypt geometry and measurement accuracy; and segmentation-based auxiliary information (villi shoulder and crypt border) was selected based on clinical insight from pathologists regarding crypt depth estimation. The rationale for these component choices is presented in Section 3 and further supported by Table 4, which shows that each component contributes measurable performance gains.
>
> > W3: A considerable number of additional terms are introduced into the loss functions. Although these terms aim to capture different aspects of the task, their relative weighting becomes increasingly problematic. For instance, the total segmentation loss is defined as Lseg = Ldice +Lce +LDTW, where the last term operates on a significantly different numerical scale. Without appropriate normalization, simply assigning all weighting factors to one may render some terms ineffective. A similar issue arises in the final training loss for Dθ. The authors state: “We simply add all terms, and do not introduce any hyperparameters in the final loss, to save on hyperparameter tuning”. However, this simplification may inadvertently diminish the impact of certain loss components
>
> We appreciate this observation. In practice, we normalize the individual loss components to ensure they operate on comparable scales. Specifically, for $L_{DTW}$, we scale by the image dimension, which brings it into the same magnitude range as $L_{dice}$​ and $L_{ce}$​. Given the large number of terms, introducing additional weighting hyperparameters would substantially expand the search space and risk overfitting on the limited validation set; our unweighted formulation promotes better generalization and avoids extensive hyperparameter tuning. Empirically, this strategy proves effective, as MeasureNet significantly outperforms the closest baselines across all key metrics. Finally, both dataset and code will be released to enable further research.
>
> >W4: W4: The comparison with other methods, referred to as baselines, is also somewhat misleading. The compared approaches are “general-purpose” methods that have not been tailored to this specific task. Consequently, the reported superior performance of the proposed model is not entirely unexpected.
>
> We introduce a new task of measurement via polyline detection, therefore pre-existing baseline are not available. To ensure a fair comparison, we adapted the closest existing approaches to this problem. We evaluate against three categories: (1) segmentation-based methods (FRNet, SegFormer, BEiT), (2) lane-detection methods (Yolino), and (3) line-detection models (MapTR, LETR), representing the most relevant modeling paradigms.
> As detailed in Supplementary:  Implementation Details, we carefully adapted each baseline to our task: for Yolino, we trained separate models for villi and crypts, reformatting polylines to match lane-detection format, and used a 32×32 grid to ensure proper endpoint alignment. For LETR and MapTR, we used the same DINO-DETR + Swin backbone as MeasureNet and matched the output configuration to predict 3-point polylines, ensuring architectural parity. These modifications were applied for both datasets (CeDeM and DeepBacs), ensuring that all baselines were given equal opportunity to perform well within the task constraints.

---

> ### Author Response · Authors · 2025-12-02
>
> > Q1:  The Earth Mover’s Distance (EMD) exhibits issues with smoothness and differentiability. Should the authors consider using a differentiable alternative, such as the Sinkhorn distance (entropic-regularized EMD)?
>
> Thank you for pointing this out. In fact, we already use a differentiable EMD implementation based on the Sinkhorn formulation. This is explicitly implemented in our code (see supplementary material, `models/dino.py`, lines 85–111). We have also updated the manuscript accordingly (marked in red at line 316): “We use a differentiable EMD based on the Sinkhorn formulation (p=2, scaling=0.9).”
>
> > Q2: How critical are the many loss terms used in training? Which terms contribute most to performance, and does tuning their weights substantially affect results?
>
> We would like to draw the reviewers’ attention to Table 4, where we ablate the contribution of each loss component EMD  Loss ($L\_{emd}$​), Length Loss ($L\_{L}​$), and Part Length Loss ($L\_{PL}​$). Both $L\_{emd}$ and $L\_{L}​$ provide substantial improvements in measurement accuracy. Specifically, incorporating $L\_{emd}​$ improves MAE–Villi from 20.18 → 18.33 and MAE–Crypt from 12.62 → 11.74 (Table 4, Rows 1–2), demonstrating the benefit of alignment with ground truth polylines. The addition of $L\_{​L}$, which guides overall length estimation, further reduces MAE–Villi and MAE–Crypt to 17.02 and 11.13, respectively.
> Finally, $L\_{LP}​$ is motivated by the observation that shift in the middle point leads to deviations in polyline curvature relative to the ground truth. Thus, $L\_{LP}​$ explicitly enhances curvature consistency. While its quantitative impact on MAE–Villi is modest (17.02 → 16.43), this is expected given the smaller number of villi with more than two points.
>
> We tried tuning the weights:
> Let us assume that $L = \lambda_{1} * L_{emd} + \lambda_{2} * L_{L} + \lambda_{3} * L_{PL}$. Overall the existing weighting method works best in our case.
>
> | $\\lambda\_{1}$ | $\\lambda\_{2}$ | $\\lambda\_{3}$ | MAE Villi  | MAE Crypt |
> | :---- | :---- | :---- | :---- | :---- |
> | 1 | 1 | 1 | **14.09** | 8.09 |
> | 2 | 1 | 1 | 16.09 | 8.08 |
> | 1 | 2 | 1 | 16.03 | 8.82 |
> | 1 | 1 | 2 | 15.50 | 8.30 |
> | 1 | 2 | 2 | 16.78 | **7.90** |
>
> > Q3: What is the LWNet reported in Table 3? This model is not described in the paper.
>
> LW-Net is a segmentation-based method primarily designed for thin vessel extraction in retinal images (“State-of-the-art retinal vessel segmentation with minimalistic models”). We have updated the manuscript (line 105, marked in red).
>
>
> > Q4: Table 4 suggests that mask augmentation and mixup drive most of the performance gains; the contribution of the three L terms is unclear. Are these loss components truly essential?
>
> The three loss components $L\_{emd}​$, $L\_{L}​$, and $L\_{PL}​$ are critical and together yield substantial improvements over the base model—from MAE–Villi 20.18 → 17.02 and MAE–Crypt 12.62 → 11.34, while also reducing MAE-ratio from 70.11 → 56.3. These losses provide polyline alignment, enforce absolute length accuracy, and improve curvature, respectively.
>
> Mask augmentation and mixup provide additional pathology insights regarding the global view of villi shoulder (starting point of crypts) and crypt border (end point of crypts). Introducing villi-shoulder and crypt-border cues (row 5\) further improves crypt prediction (MAE–crypt 11.34 → 10.23), and enabling mask mixup (row 6\) drives it further to 8.03 by increasing robustness to segmentation uncertainty. Thus, the loss terms provide the core measurement capability, while mask-based strategies provide pathology insights into the model.

---

### Author Response · Authors · 2025-12-03
**Global Comments**

We thank the reviewers for recognizing the strengths of our work. We appreciate the acknowledgement that the formulation of Celiac DIsease (CeD) analysis as a polyline-based measurement task simplifies a complex medical imaging problem and enables practical applicability. Reviewers highlighted the novelty of our weak–strong segmentation mixup approach to mitigate exposure bias, and we are encouraged that this was seen as both innovative and promising.
We also thank reviewers for emphasizing the contribution of the CeDeM dataset, with detailed villi/crypt and Villi Shoulder/Crypt Border annotations, which will be publicly released. We further appreciate the recognition that the paper presents an interesting and clinically meaningful research direction for improving celiac disease (CeD) diagnosis.

We would like to address the common concerns raised by reviewers below:
> Contribution and Novelty of the Work

The novelty lies in both the task formulation and the methodological contributions. We introduce a new task of polyline-based structural length measurement for Celiac Disease (CeD) grading, supported by a novel expert-annotated dataset (CeDeM) with villi/crypt polylines and auxiliary Villi Shoulder (VS)/Crypt Border (CB) annotations. Methodologically, we propose MeasureNet, a polyline centric detection framework with measurement driven losses (EMD, length, and part-length) explicitly designed for measurement. We further leverage pathology driven VS/CB cues to guide crypt estimation, and to the best of our knowledge we are the first to mitigate exposure bias via feature level weak and strong mixup. Moreover, we are the first to enable measurement based CeD grading, improving the current diagnostic paradigm. While developed for celiac disease, the approach generalizes to other medical imaging domains where accurate length assessment is required.

> Comparison with Baselines

We introduce a new task of measurement via polyline detection, therefore pre-existing baseline are not available. To ensure a fair comparison, we adapted the closest existing approaches to this problem. We evaluate against three categories: (1) segmentation-based methods (FRNet, SegFormer, BEiT), (2) lane-detection methods (Yolino), and (3) line-detection models (MapTR, LETR), representing the most relevant modeling paradigms.
As detailed in Supplementary:  Implementation Details (lines 807–820), we carefully adapted each baseline to our task: for Yolino, we trained separate models for villi and crypts, reformatted polylines to match lane-detection format, and used a 32×32 grid to ensure proper endpoint alignment. For LETR and MapTR, we used the same DINO-DETR + Swin backbone as MeasureNet and matched the output configuration to predict 3-point polylines, ensuring architectural parity. These modifications were applied across both datasets (CeDeM and DeepBacs), ensuring that all baselines were given every opportunity to perform well within the task constraints.

> Importance and Training of Weak–Strong Segmentation Models

In duodenum biopsy images crypts are frequently faint, partially occluded, or visually ambiguous, making them difficult to detect directly from raw tissue appearance. Therefore, similar to the reasoning used by pathologists, we use segmentation masks of villi shoulder and crypt border as auxiliary guidance which essentially provides the model where crypts begin and end.

The use of both strong and weak segmentors prevents the detector from over-relying on perfect segmentation. If the model only sees strong (high-quality) masks during training, it will fail when real-world segmentation is imperfect. By introducing a weak segmentor and performing feature-level mixup between strong and weak masks, we intentionally expose the detector to segmentation errors. This reduces exposure bias and improves robustness of crypt length estimation under varying segmentation quality. To the best of our understanding, we are the first to use mixup between strong and weak segmentation masks for reducing exposure bias. During training MeasureNet uses both strong and weak segmentation models, while during inference only the strong segmentor is used.

---

### Meta-Review · Area_Chair_YSPw · 2026-01-06

**Summary:**

In this work multiple concerns were raised, from incremental nature of the approach (TYqi, S26Q), unclear rationale (TYqi), insufficient argumentation of the algorithm design (TYqi, S26Q), misleading and non-specialized baselines (TYqi) and unfair comparisons (S26Q), presentation issues (uYFQ), limited applicability (uYFQ), lack of complexity analysis (S26Q).

Aside for one reviewer proposing borderline acceptance, the general opinion is towards reject, with two reviewers claiming high confidence.

**Reviewer Concerns:**

While most of the concerns related to the justification of the algorithm design could be cleared, together with the lack of non-specialized pipelines (it is claimed they do not exist, and the reviewer does not provide them), it is true that competing baselines are not leveraging the same powerful knowledge transfer and despite the pointing out to the supplementary material, a more in depth comparison analysis and comparison is required. Besides, the incremental nature of the proposed approach remains.

**Reviewer Scores:**

Potentially some score could have increased, but maintaining a general negative evaluation for the work.

---

### Decision · Program_Chairs · 2026-01-26

Reject